# Measurement and Calculation Techniques of Complex Permeability Applied to Mn-Zn Ferrites Based on Iterative Approximation Curve Fitting and Modified Equivalent Inductor Model

**Piotr Szczerba [1],\*** , **Slawomir Ligenza [1] and Cezary Worek [2]**

[1] Fideltronik Poland R&D Centre, Aleja Pokoju 18C, 31-564 Krakow, Poland; slawomir.ligenza@fideltronik.com
[2] Faculty of Computer Science, Electronics and Telecommunications, Institute of Electronics, AGH University of Krakow, Aleja Adama Mickiewicza 30, 30-059 Krakow, Poland; worek@agh.edu.pl
\* Correspondence: piotr.szczerba@fideltronik.com; Tel.: +48-12-618-9664

**Abstract:** In many cases, power inductors are responsible for most of the power loss, volume, and cost if applied to high-frequency power electronics applications. It is desirable to optimize their design by the proper calculation of winding and core loss. It allows faster and cheaper commercial product release, which is the key to being successful in a highly competitive market. This is only possible if existing calculation techniques and technical data given by, e.g., core manufacturers, are verified and correct; otherwise, the inductor optimization process is less precise and requires several iterations to achieve good convergence. This paper addresses existing and proposes improved measurement and calculation techniques with regard to complex permeability, one of the key quantities that define inductor behavior in the frequency domain. This is done through impedance measurement and improved definition of the equivalent inductor model. Moreover, the proposed calculation techniques fulfill the need for the simple, accurate analytical methods required in commercial designs.

**Keywords:** complex permeability; equivalent inductor model; core resistance; winding resistance; power loss; Dowell equation

## 1. Introduction

The successful inductor optimization process needs verification and improvement in the calculation of several quantities, such as winding resistance, core resistance, temperature rise, and power loss, which define inductor shape, size, and cost. These quantities depend on each other, so miscalculation in one might cause significant discrepancies in others, and thus, they must be carefully studied.

The tool, which is commonly used to do so, is a vector network analyzer (VNA). This relatively easily affordable device sweeps through frequency ranges extracting all the necessary information, which is further required in the design of an equivalent inductor model. Moreover, the VNA has one significant advantage—it allows measurement in the "off-line" mode, on the stand-alone inductor, which significantly simplifies the method, saving time and the resources needed. However, it should be remembered that this type of inductor analysis is conducted using a small-signal technique, and thus, the large-signal losses require an additional effort in the measurements, especially if the core loss is considered.

In the model, the inductor equivalent series resistance (ESR) (Figures 1 and 2) consists of embedded information on winding and core resistance, and thus an engineer must know both quantities to split the data. The winding resistance is usually calculated using the Dowell equation [1,2], while the core resistance depends mostly on the imaginary part of the complex permeability if an applied testing signal is small enough. Unfortunately, the

precise estimation of the core resistance is not always possible because the core manufacturers' data are subject to large measurement uncertainties [3]. Moreover, the real inductor itself is a complex system with many other quantities influencing permeability values. The best way to achieve good convergence in calculations is to obtain the required information directly from the complex permeability measurement, stripping the results from unwanted information.

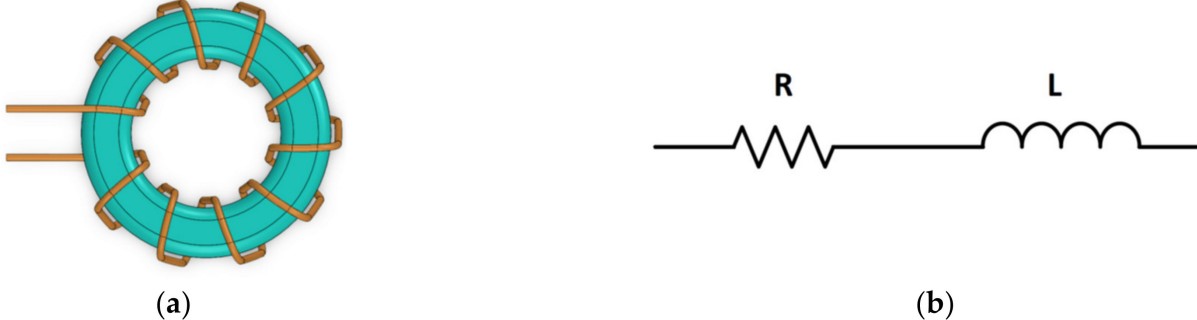

**Figure 1.** (**a**) Toroidal ferrite coil and (**b**) its equivalent simplified series model.

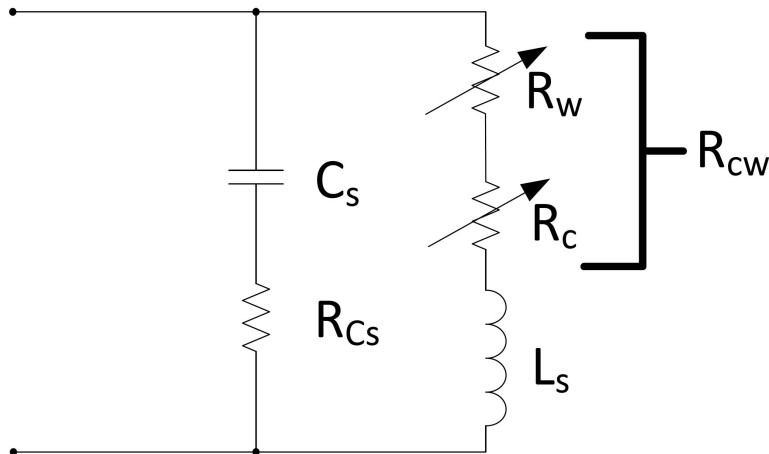

**Figure 2.** Series un-gapped high-frequency equivalent inductor model.

However, several existing complex permeability measurement and calculation techniques do not fully reveal how the complex permeability is measured and fitted, or the fitting is oversimplified [4–10] or overcomplicated [11]. The fittings are performed based on not fully complete equivalent inductor models with several fitting variables, which can take arbitrary values and do not represent the physical behavior of the inductor [12,13]. This approach cannot lead to the valid results needed for simulation programs, with integrated circuit emphasis (SPICE) commonly used in the industry to model electronic circuit behavior and quicken the time-to-market product release [14,15].

This paper discusses the existing complex permeability measurement and calculation techniques and proposes a new one based on small-signal impedance measurements, an improved equivalent inductor model, and the iterative approximation curve-fitting technique. This allows the relatively accurate estimation of complex permeability values regardless of inductor size, shape, winding structure, or frequency range. The method is simple and intuitive to process, assumes the frequency dependence of most of the inductor model components, and thus overcomes some limitations and the complexity of other methods [2,5,11–13,16]. This is significant because it might help in the development of improved inductor loss models, as well as universal simulation models (e.g., SPICE models) that capture all AC loss mechanisms, which do not yet exist, especially if we consider high-frequency power electronics applications.

## 2. Complex Permeability Verification

The verification of complex permeability values and the characteristics given by the core manufacturers are crucial to the inductor power loss and its overall optimization process [14–18]. This can be successfully performed if the inductor model is well-defined together, with all its parasitic components influencing inductor impedance measurement. It has to be considered that most of the existing inductor models used in standard power electronics applications are valid for switching frequencies up to 3 MHz, the range of which shall be significantly extended [5].

### 2.1. Definition of Inductor Lumped Model and Complex Permeability Values

Some of the research work [4,10,12] and the IEC 62044-2 standard [7] simplify the measurement procedure by assuming that the toroidal lumped inductor model (Figure 1) is only a series combination of resistance and reactance. In the case of an un-gapped toroidal core with only one layer of widely spaced windings, this approach might yield valid results; however, it is worth the additional effort to go a step further and improve the model to be even more accurate and include the remaining parasitic components in the overall calculations.

The existing high-frequency equivalent inductor models, except the simplified R-L circuit, consider the inductor stray capacitance, core resistance, leakage-inductance-related components, and the influence of inductor terminals [2,5,14,15,17–19].

Considering the power inductor, these models can be further simplified (Figure 2), removing all the parasitic components influencing the model until the multi-megahertz switching frequency is in place, where:

$C_s$　—inductor stray capacitance measured at self-resonance;
$R_{c_s}$　—inductor stray capacitance equivalent series resistance;
$L_s$　—inductor series inductance measured at low frequencies;
$R_c$　—equivalent core series resistance;
$R_w$　—windings resistance;
$R_{cw}$　—core and windings equivalent series resistance.

As reported in [14,15], the inductor lumped model should include the equivalent series resistance of the stray capacitor to make the model more realistic and avoid non-existing signal spikes if the model is placed in simulation software.

The capacitor ESR is mostly influenced by the dielectric permittivity of the winding wire coating, which is usually made of polyurethane or polyamide resin, and to a lesser extent by the dielectric permittivity of the remaining insulation materials such as kapton, mylar, and others [2]. Usually, the dielectric constant for such materials ranges from 3 to 4 [2,20–22]. Moreover, the IEC 60317-20 [23] imposes on the enameled winding wires an upper limit for the dielectric loss tangent, which is $300 \times 10^{-4}$ at 1 MHz frequency. Taking these factors into account, the inductor capacitance ESR ($R_{Cs}$) can be estimated as:

$$R_{Cs} = \frac{\tan \delta}{\omega \cdot C_s} \tag{1}$$

where:
$\tan \delta$ —dielectric loss tangent, which is assumed to be $300 \times 10^{-4}$ for copper-to-copper wire turn;
$\omega$ —angular frequency.

The impedance, the real and imaginary parts of the inductor equivalent model shown in Figure 2 can be expressed as:

$$r = \frac{\omega^4 L_s^2 C_s^2 R_{Cs} + \omega^2 C_s^2 R_{cw} R_{Cs}(R_{cw} + R_{Cs}) + R_{cw}}{(1 - \omega^2 L_s C_s)^2 + (\omega C_s (R_{cw} + R_{Cs}))^2} \tag{2}$$

$$x = \frac{\omega^3 L_s C_s \left(C_s R_{Cs}^2 - L_s\right) - \omega C_s R_{cw}^2 + \omega L_s}{\left(1 - \omega^2 L_s C_s\right)^2 + \left(\omega C_s (R_{cw} + R_{Cs})\right)^2} \tag{3}$$

where: $R_{cw} = R_w + R_c$ and thus, the effective real and imaginary parts of the complex permeability are [7]:

$$\mu_r' = \frac{l_c}{\mu_0 A_c N^2} \cdot \frac{x}{\omega} \tag{4}$$

$$\mu_r'' = \frac{l_c}{\mu_0 A_c N^2} \cdot \frac{r}{\omega} \tag{5}$$

As is self-evident, the effective complex permeability values depend on the inductor core geometry and winding structure, and thus are unique for each inductor setup [6]. So, the data given by the core manufacturers are for close-to-ideal inductors and are valid far from the inductor self-resonance; otherwise, it may only serve as a base for the initial estimation of the core magnetic permeability behavior. This assumption will be verified in the next subsection.

### 2.2. Experimental Verification of Inductor Complex Permeability Values

The experimental verification of inductor complex permeability was performed using Bode 100 VNA from Omicron Lab together with a homemade impedance adapter similar to the B-WIC test fixture at 0 dBm testing signal strength (Figure 3). The test prototypes were built with commercially available cores and winding wires. The inductors' physical parameters are shown in Table 1, while the test results are shown in Figure 4a–f and the inductors themselves in Figure 5a,b, respectively, where:

$A_c$ —core cross-section area;
$l_c$ —length of magnetic path;
$N$ —number of turns;
$Gap$ —length of the air gap;
$t$ —distance between two adjacent winding turns;
$SRF$ —inductor self-resonant frequency;
$L$ —inductance measured at approximately 10 kHz;
$C_s$ —inductor stray capacitance at self-resonant frequency;
$d$ —winding wire diameter.

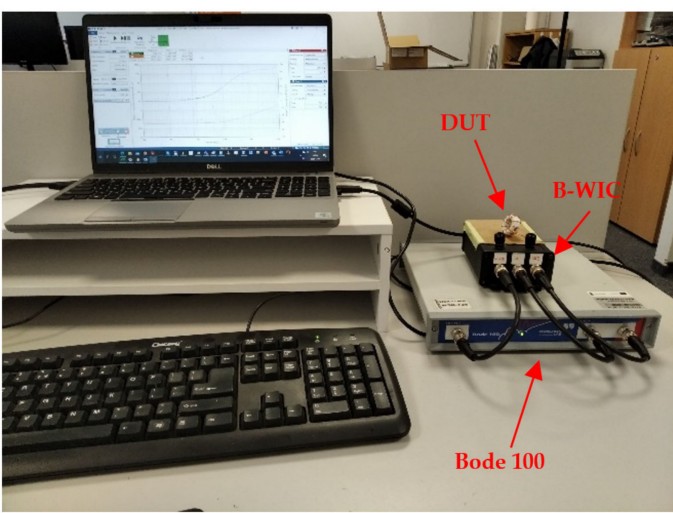

**Figure 3.** Complex permeability test-bench setup.

**Table 1.** Physical parameters of inductors under test.

| Sample No. | Core Type | Core Material | $A_c$ [mm²] | $l_c$ [mm] | N [-] | Gap [mm] | t [mm] | SRF [MHz] | L [µH] | $C_s$ [pF] | d [mm] |
|---|---|---|---|---|---|---|---|---|---|---|---|
| Sample 1 | TN 10/6/4 | 3C90 | 7.8 | 24.1 | 10 | N/A | 2.5 | 9.907 | 103.78 | 2.48 | 0.28 |
| Sample 2 | TN 20/10/7 | 3C90 | 33.6 | 43.6 | 10 | N/A | 4.7 | 2.814 | 234.17 | 13.65 | 0.75 |
| Sample 3 | TN 25/15/10 | 3C90 | 48.9 | 60.2 | 10 | N/A | 6.3 | 2.954 | 237.23 | 12.49 | 0.75 |
| Sample 4 | TX 58/41/18 | 3E25 | 152 | 152 | 10 | N/A | 15.6 | 1.262 | 574.60 | 27.64 | 0.75 |
| Sample 5 | ETD 44/22/15 | 3C90 | 173 | 103 | 10 | N/A | 3.2 | 2.365 | 331.23 | 13.67 | 0.75 |
| Sample 6 | ETD 44/22/15 | 3C90 | 173 | 103 | 10 | 1.3 | 3.2 | 27.663 | 21.17 | 1.56 | 0.75 |

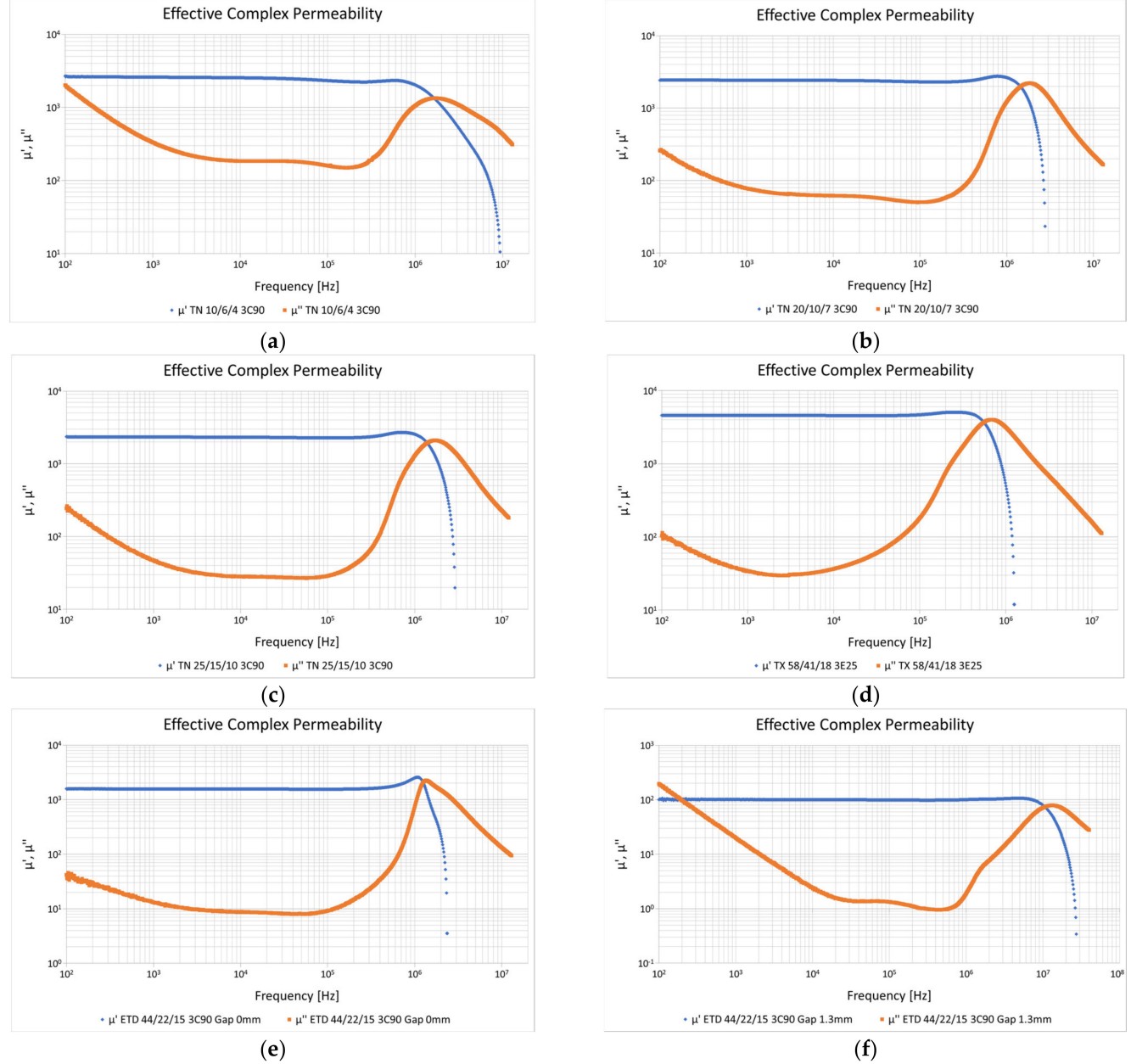

**Figure 4.** Effective complex permeability plots: blue—real part of complex permeability; orange—imaginary part of complex permeability; (**a**) sample 1; (**b**) sample 2; (**c**) sample 3; (**d**) sample 4; (**e**) sample 5; (**f**) sample 6.

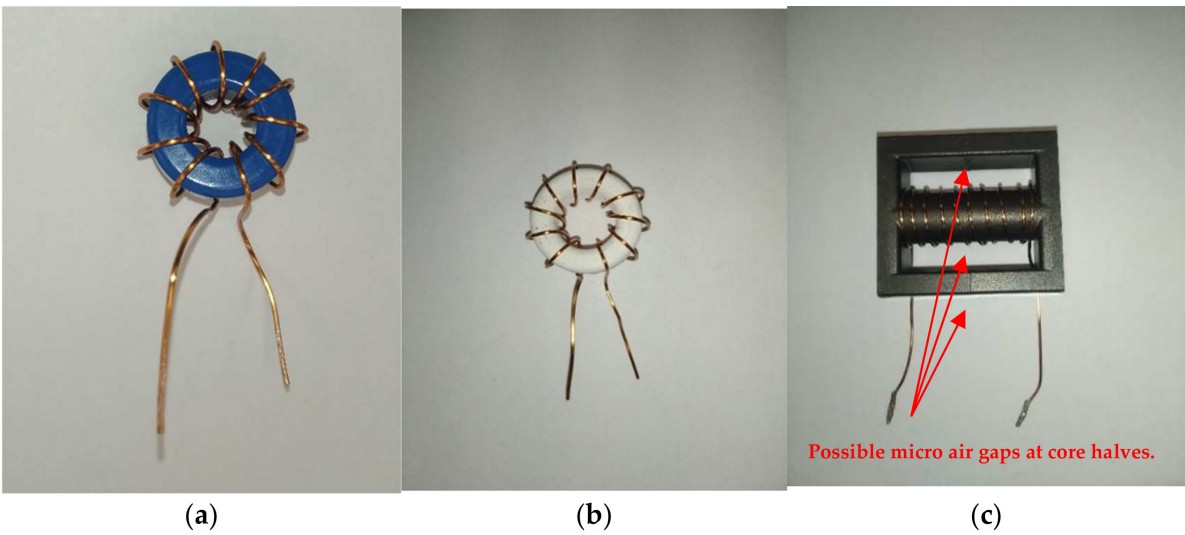

| (**a**) | (**b**) | (**c**) |

**Figure 5.** (**a**) Sample 2 used in verification of complex permeability characteristics; (**b**) sample 3 used in verification of complex permeability characteristics; (**c**) sample 5 with possible micro air gaps affecting complex permeability values.

The testing signal strength as per IEC 62044-2 was chosen so as not to exceed $B_{max} = 0.25$ mT.

As shown in Figure 4a–f, the core shape and the windings' structure impact the complex permeability values as predicted. This makes the permeability characteristics unique for each inductor, especially near the inductor's self-resonant frequency.

The imaginary part of the effective complex permeability characteristics bends upwards at low frequencies. This phenomenon is mostly due to the resistance of the inductor windings and, to a lesser extent, the remaining parasitic RLC elements of which the physical inductor is made (Figure 2).

The effective complex permeability model itself represents the physical behavior of a real inductor, where the resistance of the winding wires and other parasites must be included. Therefore, the bending might be removed from the measurement, e.g., by a curve fitting [4], because it represents a physical change from the dominance of the core loss to the winding loss.

The possible cracks in the core material or small air gaps not visible to the naked eye, which, e.g., exist at the joints of core halves, might significantly decrease the complex permeability, and thus the inductance of an inductor itself (Figures 4e and 5c).

The above is even more self-evident if the gapped inductor is considered. The influence of the gap, and thus its resistance and the resistance of the fringing field, dominates, significantly decreasing the values of the real and imaginary parts of the complex permeability characteristics (Figure 4f).

The low-frequency bending phenomenon will be investigated in the next subsection.

### 2.2.1. Mathematical Verification of Low-Frequency Effective Complex Permeability

As shown, the measured low-frequency imaginary part of complex permeability bends upwards to the higher values. Mathematically, this phenomenon might be explained by the fact that when the frequency decreases toward zero (Equation (2)), the real part of the inductor impedance tends toward the frequency-independent finite value, which is the windings' DC resistance $(R_{wDC})$. This can be written as:

$$\lim_{\omega \to 0} r = R_{wDC} \tag{6}$$

while at the same time, the imaginary part of the effective complex permeability tends toward infinity:

$$\lim_{\omega \to 0} \mu_r'' = \frac{l_c}{\mu_0 A_c N^2} \cdot \frac{r}{\omega} = \frac{l_c}{\mu_0 A_c N^2} \cdot \frac{R_{wDC}}{\omega} \to \infty \tag{7}$$

The bending indeed exists and is caused by a natural phenomenon, which comes from the inductor lumped model and is a part of the effective complex permeability model itself.

On the other hand, the above phenomenon does not exist if we consider the real part of complex permeability. In this case, the imaginary part of the inductor impedance tends toward zero

$$\lim_{\omega \to 0} x = 0 \tag{8}$$

while the real part of the complex permeability tends toward a high but finite value, which depends on the inductor's lumped parameters and can be expressed as follows:

$$\lim_{\omega \to 0} \mu_r' = \frac{l_c}{\mu_0 A_c N^2} \cdot \frac{x}{\omega} = \frac{l_c}{\mu_0 A_c N^2} \cdot \left( -C_s R_{cw}^2 + L_s \right) \approx \frac{l_c}{\mu_0 A_c N^2} \cdot L_s \tag{9}$$

Based on the measurement results from Section 2.2 and mathematical explanation of the low-frequency bending phenomenon, the new iterative approximation curve-fitting technique together with the verification of complex permeability characteristics will be presented next.

2.2.2. Verification of Effective Complex Permeability Characteristics Based on Inductor Equivalent Model and Iterative Approximation Curve Fitting

The measured complex permeability curves can be verified by the iterative approximation curve-fitting technique based on a series un-gapped equivalent inductor model (Figure 2). The inductors chosen to do the fitting are sample 2 and sample 3 with ferrite core made of 3C90 material from Ferroxcube and wound with 10 turns of widely spaced, 0.75 mm-diameter, enameled copper wire (Figure 5a,b). The average space between adjacent winding wires was assumed to be 4.7 mm for sample 2 and 6.3 mm for sample 3 (Table 1). During the fitting process, the samples' impedance was measured from 100 Hz to 13 MHz using Bode 100 VNA (Figure 6a–d). The inductance measurement was taken at a frequency of about 10 kHz, which lies within the plateau range of the measured characteristics. It is suspected that most of the inductor stray capacitance is between the wire turns and between the wire turns and the core (Figure 7). Therefore, the dielectric loss tangent of the capacitor ESR is a complex case influenced by the dielectric permittivity of several materials such as enameled copper wire coating, the air between the wire and the core, the core coating, and the core material itself.

Unfortunately, too little research has been conducted on the electrical parameters of such materials, so not many loss tangent characteristics versus frequency are commonly available [6,24]. It is assumed that most of the capacitor ESR loss tangent is due to the ratio of the core imaginary and the real part of complex permittivity, which might sharply increase at high frequencies due to the ferrite material structure itself, the core shape, and its dimensional resonance [6,25]. Based on the complex permeability curve-fitting results, it was assumed that the loss tangent is equal to 4 across the entire frequency range. This assumption does not influence the low-frequency calculations but allows reasonably good curve-fitting results close to and above the inductor self-resonance.

The approximation Itself assumes that the Inductor windings' resistance changes along the frequency range according to the standard Dowell equation [1,2,17]. This assumption yields accurate results, especially if the inductor has no distributed or discrete air gaps and is made of one layer of widely spaced winding turns.

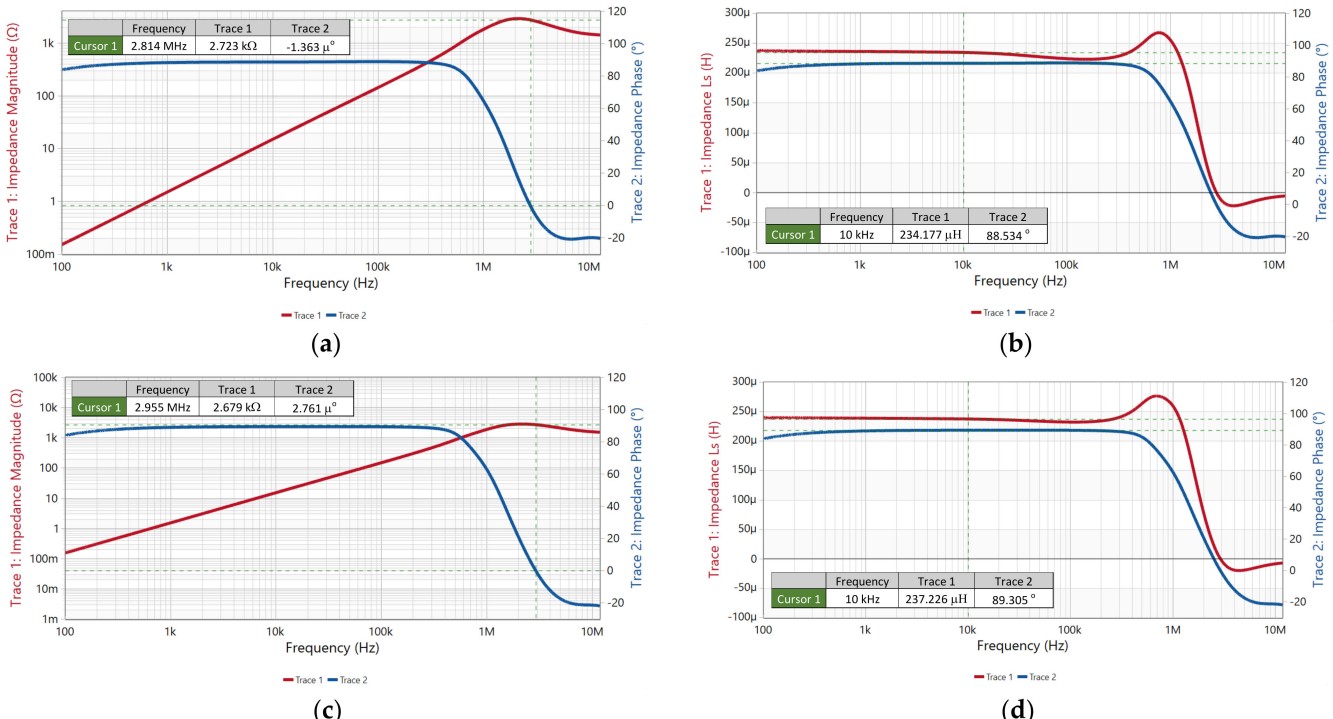

**Figure 6.** (**a**) Sample 2 impedance plot: red—impedance magnitude; blue—impedance phase; (**b**) sample 2 inductance plot: red—inductance value; blue—impedance phase; (**c**) sample 3 impedance plot: red—impedance magnitude; blue—impedance phase; red—impedance phase; (**d**) sample 3 inductance plot: red—inductance value, blue—impedance phase.

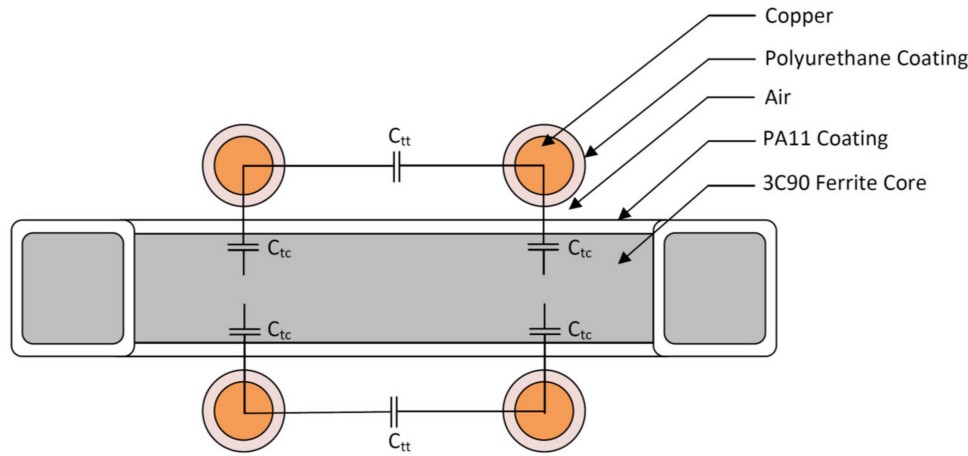

**Figure 7.** Sample 2 and 3 cross-section with highlighted turn-to-turn and turn-to-core capacitance.

The extraction assumes that the measured complex permeability values follow Equations (4) and (5) where $r$ and $x$ come from the impedance measurement. Then, the obtained results are compared with estimated ones, which are also calculated with the help of Equations (4) and (5); however, this time $r$ and $x$ come from Equations (2) and (3), namely:

$$\mu'_{r_{estimated}} = \mu'_{r_{measured}} + \xi_{\mu'_r} \tag{10}$$

$$\mu''_{r_{estimated}} = \mu''_{r_{measured}} + \xi_{\mu''_r} \tag{11}$$

$$\xi_{\mu'_r} = \mu'_{r_{measured}} - \mu'_{r_{estimated}} \tag{12}$$

$$\xi_{\mu_r''} = \mu_{r_{measured}}'' - \mu_{r_{estimated}}'' \tag{13}$$

where:

$\mu_{r_{measured}}'$ —real part of effective complex permeability (the measured characteristic);

$\mu_{r_{measured}}''$ —imaginary part of effective complex permeability (the measured characteristic);

$\mu_{r_{estimated}}'$ —estimated real part of complex permeability, using Equations (3) and (4);

$\mu_{r_{estimated}}''$ —estimated imaginary part of complex permeability, using Equations (2) and (5);

$\xi_{\mu_r'}$ —shift between measured and estimated real part of complex permeability characteristic;

$\xi_{\mu_r''}$ —shift between measured and estimated imaginary part of complex permeability characteristic.

The fitting variable is the imaginary part of the complex permeability, which is part of the core series resistance $(R_c)$, and is expressed as follows:

$$R_c = \frac{\mu_{r_{fitted}}'' \mu_0 N^2 A_c \omega}{l_c} \tag{14}$$

The $\mu_{r_{fitted}}''$ is the value, which is stripped away from the influence of the parasitics and represents the actual core loss. Its value can be obtained during the iterative sweep (in this case, sweep from 1 to 2400 with a step of 1) when the absolute value of the shift given by Equation (13) is minimal:

$$\xi_{\mu_{r_{min}}''} = MIN \left| \mu_{r_{measured}}'' - \mu_{r_{estimated}}'' \right| \tag{15}$$

To obtain the best fitting results, the series inductance $L_s$ was assumed to be a variable one, which changes along the frequency range following the changes in the measured real part of complex permeability, as such:

$$L_s = \frac{\mu_{r_{measured}}' \mu_0 N^2 A_c}{l_c} \tag{16}$$

As can be seen from the plots (Figures 8–11), the fitted values of the imaginary part of the complex permeability flatten out at low frequency, being stripped away from the influence of winding resistance, as was predicted in [4]. The estimated and measured complex permeability characteristics align at higher frequencies, suggesting that they are closely fitted. The relative fitting error between $\mu_{r_{measured}}''$ and $\mu_{r_{estimated}}''$ does not exceed 2% for the loss tangent greater than 2 for both of the testing samples. This proves that the inductor model matches reality and the method proposed seems to be correct.

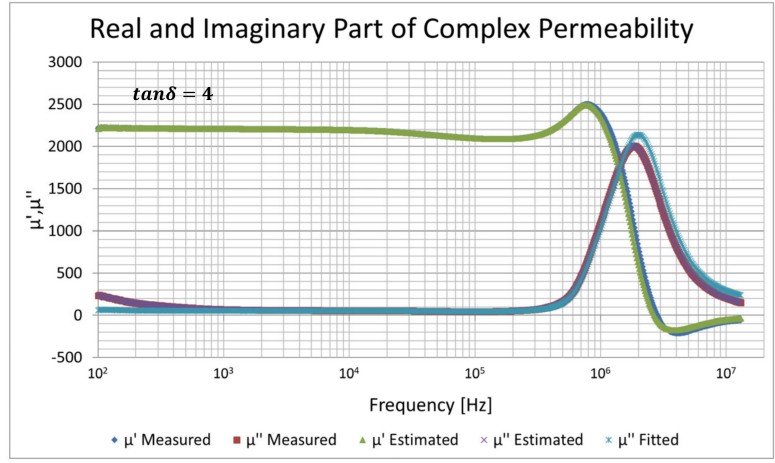

**Figure 8.** Sample 2 real and imaginary parts of complex permeability: dark blue—real part of measured characteristic; green—real part of estimated characteristic; dark red—imaginary part of

measured characteristic; purple—imaginary part of estimated characteristic; light blue—imaginary part of fitted characteristic.

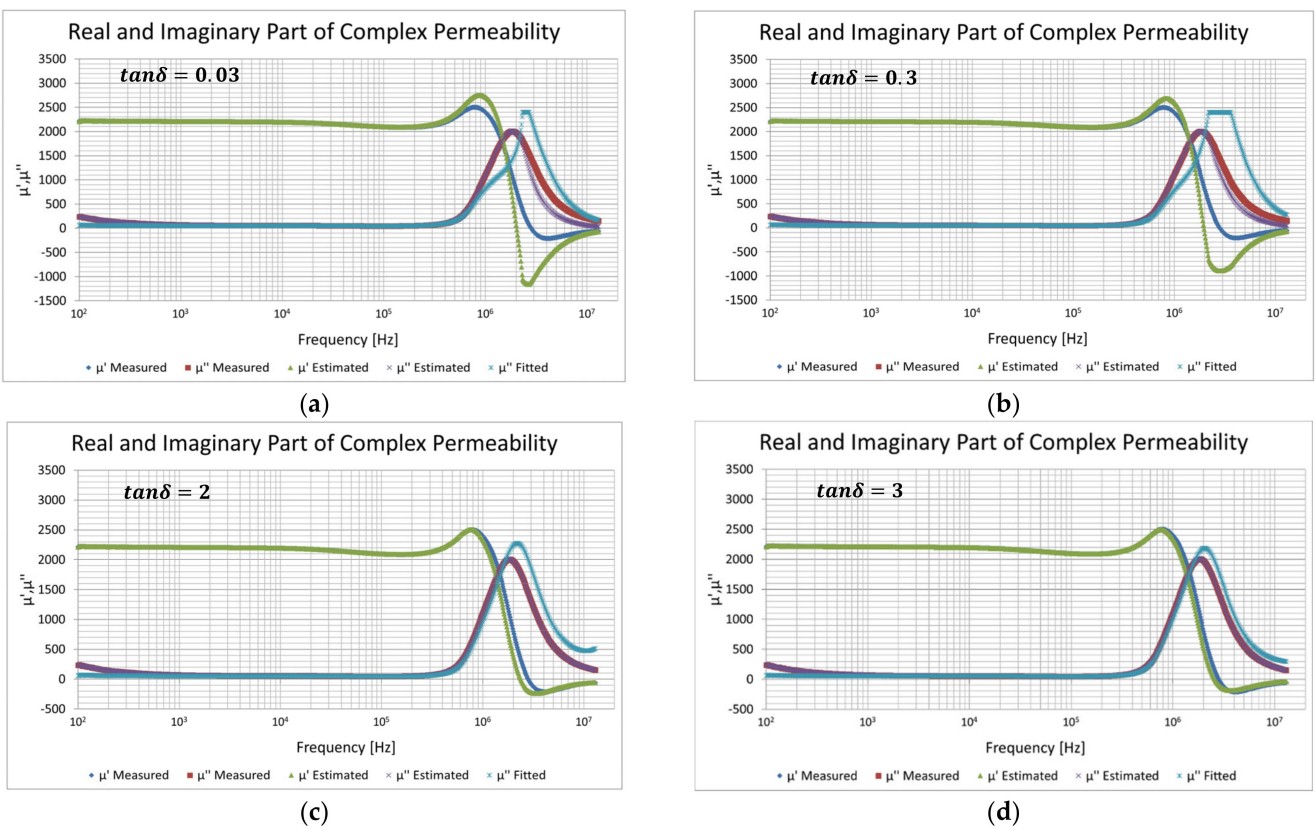

(**a**)　　　　　　　　　　　　　　　　　　　　　　(**b**)

(**c**)　　　　　　　　　　　　　　　　　　　　　　(**d**)

**Figure 9.** Sample 2 fitting convergence depending on loss tangent value: dark blue—real part of measured characteristic; green—real part of estimated characteristic; dark red—imaginary part of measured characteristic; purple—imaginary part of estimated characteristic; light blue—imaginary part of fitted characteristic; (**a**) loss tangent 0.03; (**b**) loss tangent 0.3; (**c**) loss tangent 2; (**d**) loss tangent 3.

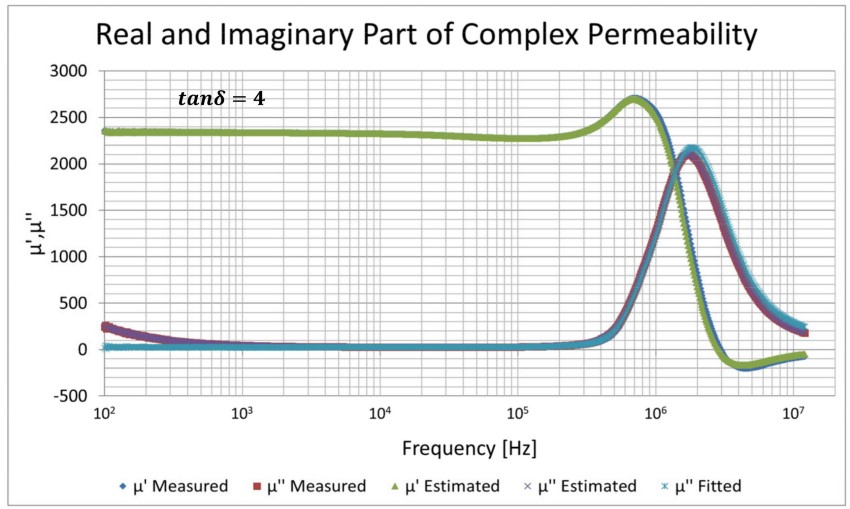

**Figure 10.** Sample 3 real and imaginary parts of complex permeability: dark blue—real part of measured characteristic; green—real part of estimated characteristic; dark red—imaginary part of measured characteristic; purple—imaginary part of estimated characteristic; light blue—imaginary part of fitted characteristic.

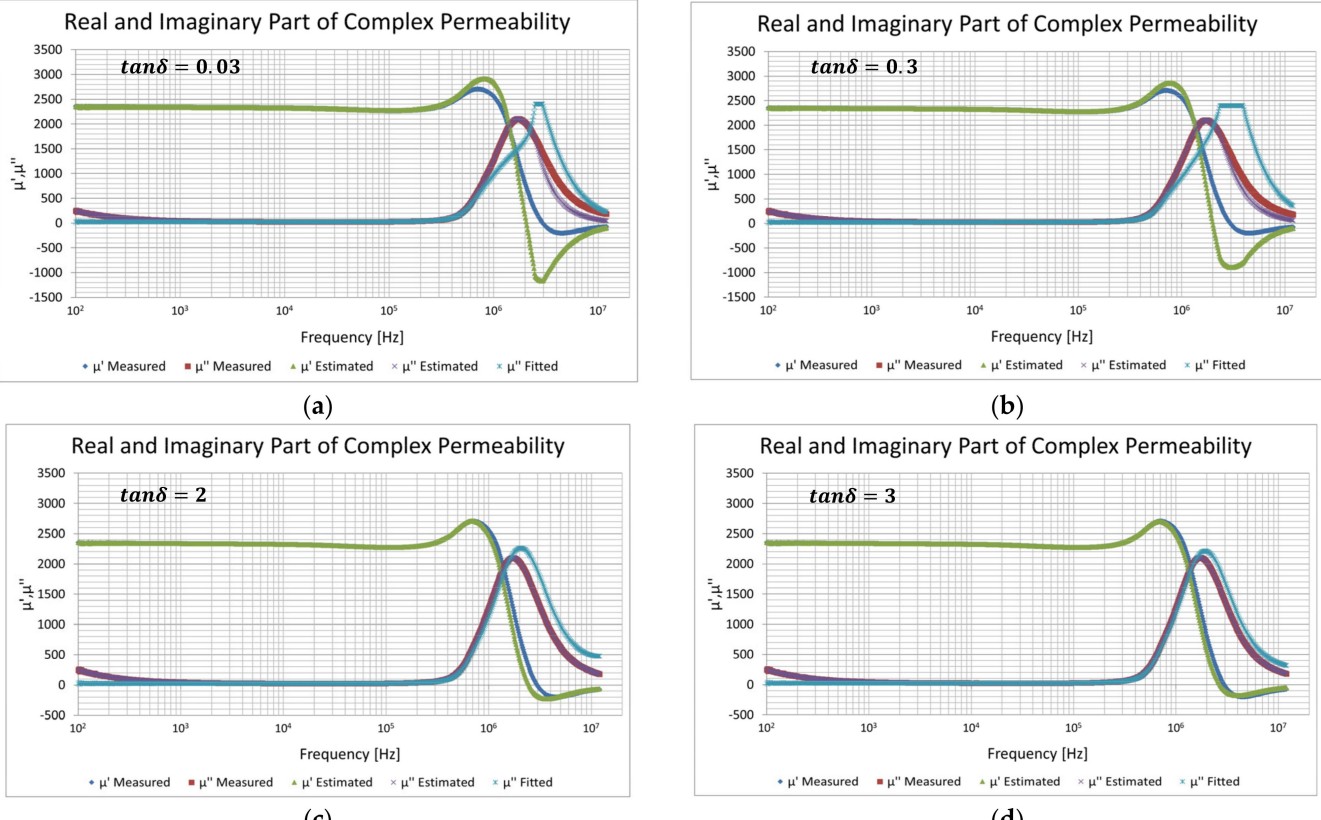

**Figure 11.** Sample 3 fitting convergence depending on loss tangent value: dark blue—real part of measured characteristic; green—real part of estimated characteristic; dark red—imaginary part of measured characteristic; purple—imaginary part of estimated characteristic; light blue—imaginary part of fitted characteristic; (**a**) loss tangent 0.03; (**b**) loss tangent 0.3; (**c**) loss tangent 2; (**d**) loss tangent 3.

The method heavily depends on the inductor stray capacitor ESR $(R_{Cs})$ value (Figures 9a–d and 11a–d). Unfortunately, there are not sufficient data available, which would prove or disprove this phenomenon and would clearly estimate the ESR value along the frequency range. The approach to limit its range and estimate its value will be shown in the next subsection.

### 2.2.3. Estimation of Inductor Stray Capacitor ESR Value

The ESR of the inductor stray capacitor $(R_{Cs})$ depends on the dielectric loss tangent (Equation (1)) of several materials (Figure 7). As suspected, one of the biggest contributors to the ESR is the loss tangent of the core coating and the core itself. In order to verify this, a homemade testing fixture has been developed (Figure 12a,b).

The test fixture has a set of two electrodes placed on adjustable arms to fit in between toroidal cores with an external diameter of up to 25 mm. During the test, a sample is placed between electrodes and its admittance is measured.

In the method presented in this paper, the calculations are performed with an inductor series equivalent model. In addition, the stray capacitance within the model also has the ESR in series. The relationship between admittance, complex permittivity, and the loss tangent calculations shall also be described by a series capacitor model, which can be expressed as follows:

$$tan\delta = \frac{\omega\epsilon_r'' + \sigma}{\omega\epsilon_r'} \tag{17}$$

$$Y_s = G_s + j\omega C_s \tag{18}$$

$$\epsilon_r = \epsilon_r' - j\epsilon_r'' \tag{19}$$

$$\epsilon_r' = \frac{C_s d_s}{\epsilon_0 A_s} \tag{20}$$

$$\epsilon_r'' = \frac{G_s d_s}{\omega \epsilon_0 A_s} = \frac{\sigma}{\omega \epsilon_0} \tag{21}$$

where:

$\omega$ —angular frequency;
$G_s$ —measured series conductance;
$C_s$ —measured series capacitance;
$\epsilon_r$ —series relative complex permittivity;
$\epsilon_r'$ —real part of series complex permittivity;
$\epsilon_r''$ —imaginary part of series complex permittivity;
$\sigma$ —material conductivity;
$d_s$ —sample thickness;
$A_s$ —sample cross-section.

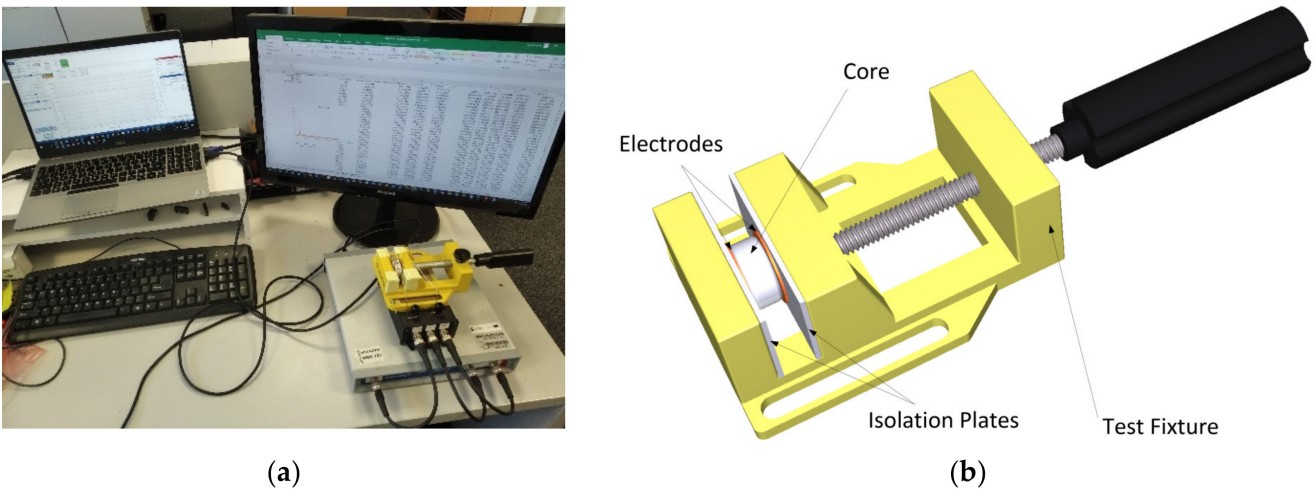

(**a**)                                                                 (**b**)

**Figure 12.** (**a**) Complex permittivity test-bench; (**b**) complex permittivity test fixture.

During the measurement, the inductor core together with the coating is placed between electrodes, and its impedance is measured by a Bode 100 VNA. Before measurement, to minimize any possible measurement error, the short, open, and load compensation is performed. Fortunately, the loss tangent is not directly dependent on the sample size and shape except for the effect of dimensional resonance (Equations (17)–(21)). This phenomenon [6,25] in this particular case depends on more than the influence of the granular structure of the 3C90 ferrite, and this will be discussed later in this paper.

The test results show (Figure 13a,b) that the loss tangent of the polyamide 11 (PA11) dielectric coating to 3C90 ferrite core for both samples (sample 2 and sample 3) can reach up to 0.048 at the inductor self-resonant frequency and up to 0.16 at 13 MHz, which is the end of the complex permeability measurement range. The 0.16 value sets the lower limit for the sought loss tangent. This value, even though expected, is surprisingly low and far from the assumed 4. In fact, a toroidal inductor has a relatively complex structure, and thus the loss tangent mechanism might originate somewhere else.

One of the ways to figure this out is to look at the inductor's behavior at and beyond its self-resonant frequency, where the inductor becomes a capacitor, and the complex permittivity defines its value and the value of the loss tangent with regard to the whole inductor.

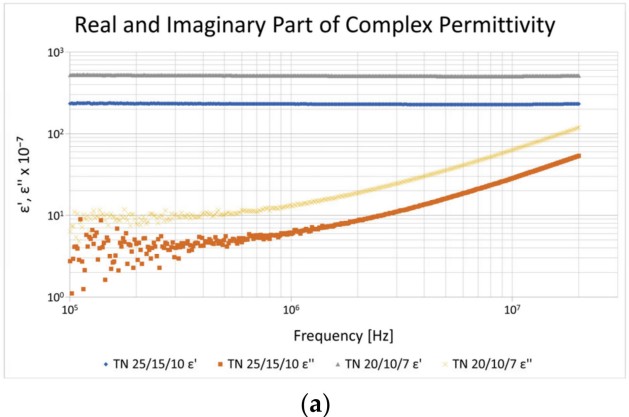

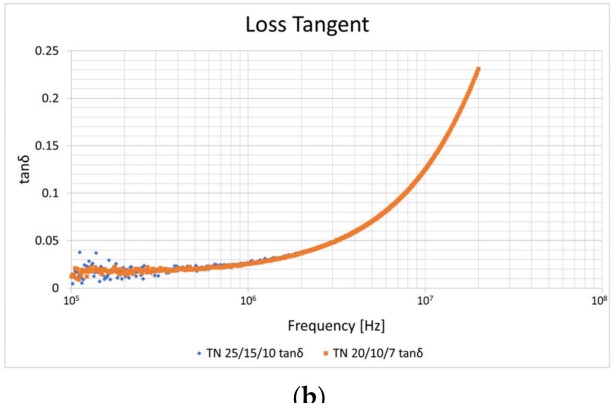

(**a**)                                    (**b**)

**Figure 13.** Coating-to-core permittivity and loss tangent characteristics. (**a**) Real and imaginary parts of complex permittivity: grey—real part of sample 2; yellow—imaginary part of sample 2; blue—real part of sample 3; orange—imaginary part of sample 3; (**b**) loss tangent: orange—sample 2; blue—sample 3.

The impedance sweeps of sample 2 and sample 3 were again performed from 100 Hz up to 40 MHz and the inductor admittance, complex permittivity and the loss tangent was calculated using Equations (17)–(21) (Figure 14a,d). In this case, we do not know the exact value of the capacitor thickness and cross-section due to the non-obvious capacitor structure, which is, in fact, a structure of a toroidal inductor. The loss tangent does not depend very much on the sample thickness and cross-section, and thus Equation (17) can be simplified to:

$$tan\delta \approx \frac{\epsilon_r''}{\epsilon_r'} = \frac{G_s}{\omega C_s} \tag{22}$$

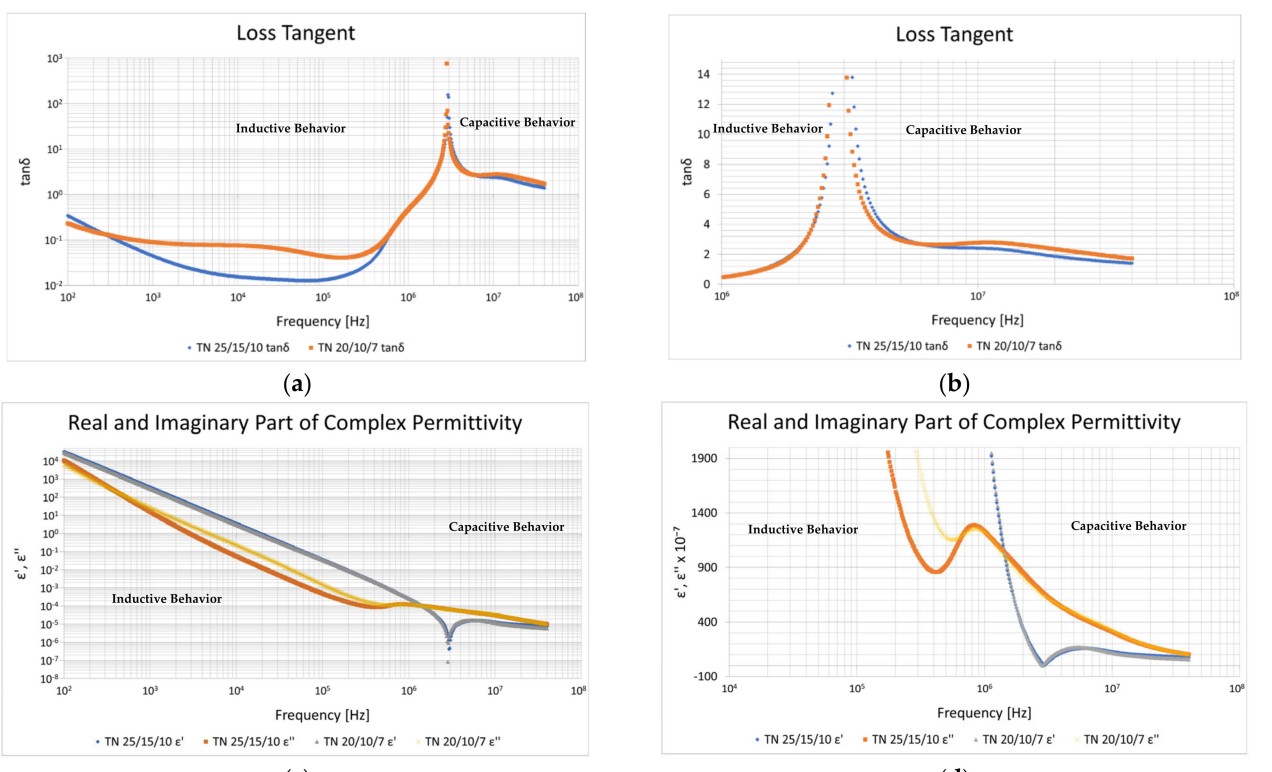

(**c**)                                    (**d**)

**Figure 14.** Sample 2 and sample 3 loss tangent and complex permittivity characteristics. (**a**) Loss tangent: orange—sample 2; blue—sample 3; (**b**) loss tangent—zoom at self-resonant frequency:

orange—sample 2; blue—sample 3; (**c**) real and imaginary parts of complex permittivity: gray—real part of sample 2; yellow—imaginary part of sample 2; blue—real part of sample 2; orange—imaginary part of sample 3; (**d**) real and imaginary parts of complex permittivity zoom at self-resonant frequency: gray—real part of sample 2; yellow—imaginary part of sample 2; blue—real part of sample 2; orange—imaginary part of sample 3.

The characteristics show an exponential decay of the loss tangent ranging from approximately 753 for sample 2 and 156 for sample 3 at the inductor self-resonant frequency to approximately 2.7 for sample 2 and 2.3 for sample 3 at 13 MHz. The sought loss tangent value lies somewhere between these extremes, and if we include the value from the previous measurement, then the possible loss tangent range starts from 0.16 to 753 for sample 2 and 0.16 to 156 for sample 3.

Because the equivalent inductor model beyond the self-resonant frequency consists of two parallel branches of a series R-C connection, it is difficult to distinguish the exact value of the inductor stray capacitor ESR ($R_{Cs}$) directly from the measurement of the overall capacitive inductor admittance/impedance. The loss tangent range does not give us an exact answer to what the loss tangent is, but it narrows the possible choice.

The calculations and the test-bench results show that the best fitting results are with the peak loss tangent values obtained at the inductor self-resonance. It suggests that the stray capacitance has only a negligible impact on the complex permeability values in the method presented herein.

A better understanding of the origin of the inductor stray capacitance and its ESR requires an investigation of the core resonance phenomena and their influence on the complex permeability values, which will be shown next.

### 2.2.4. Influence of Core Natural and Dimensional Resonance on Complex Permeability

In general, there are at least three factors that contribute to the resonance in Mn-Zn ferrites [26–28], namely:

- resonance due to the windings' stray capacitance and the inductor self-inductance;
- the windings behave as a distributed constant line;
- inherent characteristics of the magnetic material.

As stated in previous paragraphs, the inductor stray capacitance and its ESR, which are assumed to be mostly related to the windings, seem not to impact the inductor complex permeability characteristics, and the stray capacitor is somehow excluded from the overall calculations. It contradicts what one might expect and suggests that the inductor resonance originates somewhere else.

Moreover, the phenomenon when the winding behaves as a distributed constant line is mostly profound in inductors with multiple turns at multi-megahertz frequencies in the form of self-repeating resonance. This phenomenon is not visible in Figure 6a–d, so it shall also be excluded.

The resonance phenomenon, which still pertains, is the resonance due to the inherent characteristics of the magnetic material. In this case there are two phenomena:

- natural resonance;
- dimensional resonance.

Natural resonance happens to the ferrites with high magnetic permeability due to the resonance of magnetization rotation under the action of the anisotropy field [26,27]. Above the resonance, the real part of complex permeability drops along the line called Snoke's limit.

Snoke's limit can be calculated as follows [26,27]:

$$f \cdot \mu_r = \frac{v \cdot B_s}{3 \cdot \pi \cdot \mu_0} \tag{23}$$

$$v = g_e \cdot q_e \cdot \frac{\mu_0}{2 \cdot m_e} \tag{24}$$

where:

$f\mu_r$ —Snoke's limit in [MHz];
$f$ —switching frequency;
$\mu_r$ —relative magnetic permeability;
$v$ —gyromagnetic constant;
$B_s$ —assumed magnetic saturation level of ferrite material;
$\mu_0$ —magnetic permeability of free space;
$g_e$ —electron g-factor;
$q_e$ —electron charge;
$m_e$ —electron mass.

As shown in Figure 15a,b, the change in the real part of the complex permeability characteristics for sample 2 and sample 3 is preceded by the permeability increase and then followed by its sharp drop. The drop happens significantly before Snoke's limit, which suggests that the magnetic resonance cannot be attributed to the natural resonance but rather to the second phenomenon, which is related to the shape and dimensions of the inductor core.

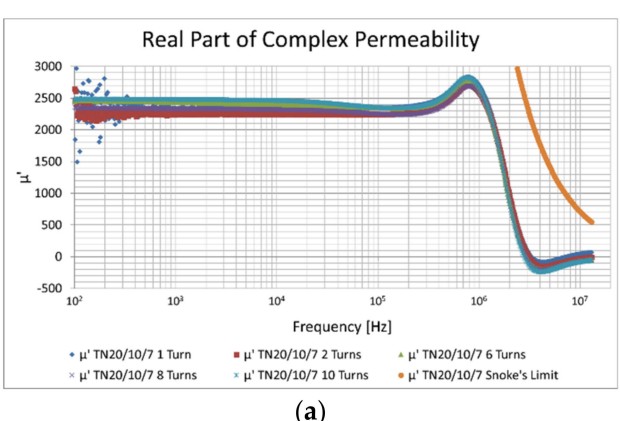

(**a**)

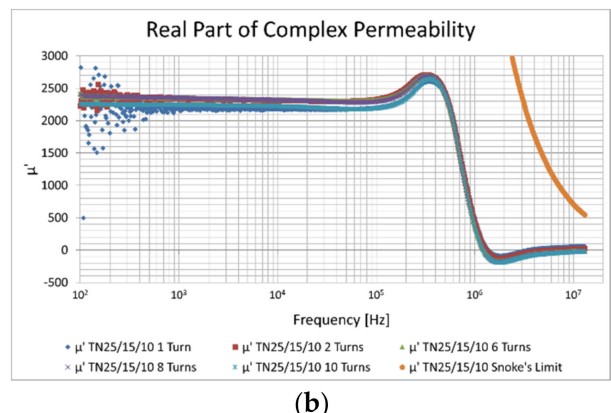

(**b**)

**Figure 15.** Sample 2 and sample 3 real part of complex permeability characteristics at different number of turns and Snoke's limit; (**a**) sample 2: blue—1 turn; red—2 turns; green—6 turns; violet—8 turns; light blue—10 turns; orange—Snoke's limit; (**b**) sample 3: blue—1 turn; red—2 turns; green—6 turns; violet—8 turns; light blue—10 turns; orange—Snoke's limit.

For the sake of explanation, in the steady state at low frequencies, the field inside the core is uniform and in phase with the excitation field provided by the windings, which represents the quasi-static field assumption. However, if the frequency increases, then the field tends to be concentrated at the surface of the core, and this is a well-known skin effect that also applies to conducting materials such as ferrites.

Because of the combined effect of the high-resistivity phase of the ferrite grains and the high permittivity and high permeability at the same time, the excitation field will propagate through the core at a different velocity than in free space. This reduced velocity can be expressed as a product of the wavelength, which is:

$$\lambda = \frac{c}{f\sqrt{\mu_r \epsilon_r}} \tag{25}$$

where:

$c$ —speed of light;
$f$ —switching frequency;
$\mu_r$ —ferrite relative permeability;
$\epsilon_r$ —ferrite relative permittivity.

In this case, the shorter wavelength causes phase displacement between the magnetic field inside the core and the magnetic field on its surface, and the quasi-static field assumption does not apply anymore. It causes the net flux linking the windings to no longer be a function of the core cross-sectional area. Moreover, if the dimensions of the core are integer multiples of the wavelength, then the electromagnetic wave will resonate inside the core and the crest of the standing wave will trigger the dimensional resonance. If this happens, then the net flux linkage of the windings is zero, resulting in no apparent inductance [26–28].

As mentioned, the phenomenon of dimensional resonance is clearly visible in Figure 15a,b, where the real part of the complex permeability increases just below the resonant frequency and then sharply drops above the resonance significantly before Snoke's limit. The characteristics show that the resonant frequency is almost independent of the number of the winding turns, suggesting that the inductor stray capacitance and its ESR in the given configurations is mostly due to the capacitance and the ESR of the core, which will be discussed in the next paragraph.

2.2.5. Estimation of Inductor Stray Capacitance Value

An estimation of the core and the winding capacitances and their ESRs would improve the lumped inductor model and help in the more precise extraction of the complex permeability characteristics.

The theoretical estimation of the windings' capacitance has been made using the method given in [2] (Table 2), namely:

$$C_{tt} = \frac{2\epsilon_0 MLT}{\sqrt{\left(\frac{1}{\epsilon_r}\ln\frac{d_0}{d_i}+1\right)^2 - 1}}\arctan\left(\sqrt{1 + \frac{2}{\frac{1}{\epsilon_r}\ln\frac{d_0}{d_i}}}\right) \tag{26}$$

$$C_{s_{nc}} = \frac{C_{tt}}{N-1} \tag{27}$$

$$C_{1,N} = C_{tt} + \frac{C_{1(N-2)}C_{tt}}{2C_{1(N-2)} + C_{tt}} \tag{28}$$

$$C_{s_{wc}} \approx 1.366 C_{tt} \; for \; N \geq 10 \tag{29}$$

$$C_{s_{dc}} \approx 1.366 C_{s_{nc}} = 1.366 \frac{C_{tt}}{N-1} \; for \; N \geq 10 \tag{30}$$

where:
$C_{tt}$ —turn-to-turn capacitance of single layer inductor;
$N$ —number of turns;
$MLT$ —mean length of turn;
$\epsilon_r$ —relative permittivity;
$\epsilon_0$ —permittivity of free space;
$d_0$ —winding wire diameter with coating;
$d_i$ —bare winding wire diameter;
$C_{snc}$ —inductor stray capacitance without a core;
$C_{1,N}$ —inductor stray capacitance with a core and N turns;
$C_{s_{wc}}$ —approximated inductor stray capacitance with a core;
$C_{s_{dc}}$ —approximated inductor stray capacitance with a dummy core;
$SRF_{dc}$ —self-resonant frequency of an inductor with a dummy core;
$C_{s_{dc(meas)}}$ —measured inductor stray capacitance with a dummy core;
$L_{dc}$ —inductance of an inductor with a dummy core.

**Table 2.** Dummy-core-based sample 2 and sample 3 parameters.

| Sample No. | MLT [mm] | $d_i$ [mm] | $d_0$ [mm] | $\varepsilon_r$ [-] | N [-] | $L_{dc}$ [nH] | $SRF_{dc}$ [MHz] | $C_{tt}$ [pF] | $C_{swc}$ [pF] | $C_{snc}$ [pF] | $C_{sdc}$ [pF] | $C_{sdc(meas)}$ [pF] |
|---|---|---|---|---|---|---|---|---|---|---|---|---|
| Sample 2 | 34.5 | 0.75 | 0.775 | 4 | 10 | 211 | 305 | 7.18 | 9.80 | 0.80 | 1.09 | 1.29 |
| Sample 3 | 42.0 | 0.75 | 0.775 | 4 | 10 | 328 | 236 | 8.74 | 11.93 | 0.97 | 1.33 | 1.39 |

This method has been verified two-fold: empirically by laboratory measurements and by the finite element method (FEM) analysis.

In the laboratory measurement, the inductors' windings are wound on acrylonitrile butadiene styrene (ABS) dummy cores with an assumed relative permeability ($\mu_r$) equal to 1. This approach allows for keeping the turns properly structured without influencing the ferrite core and with negligible impact on the windings' capacitance by the non-magnetic dummy core.

At the beginning, the windings' self-inductance is measured at approximately 5 MHz, which is the plateau region of the impedance phase and close to the 90-degree phase shift (Figure 16a,b). Then, the windings' self-resonant frequency is captured using HP8753E VNA (Figure 16c,d) with the one-port method. The measured winding capacitance is simply a product of the inductor's self-resonant frequency and the value of the inductance, which is:

$$C_{sdc(meas)} = \frac{1}{(2\pi SRF_{dc})^2 \cdot L_{dc}} \tag{31}$$

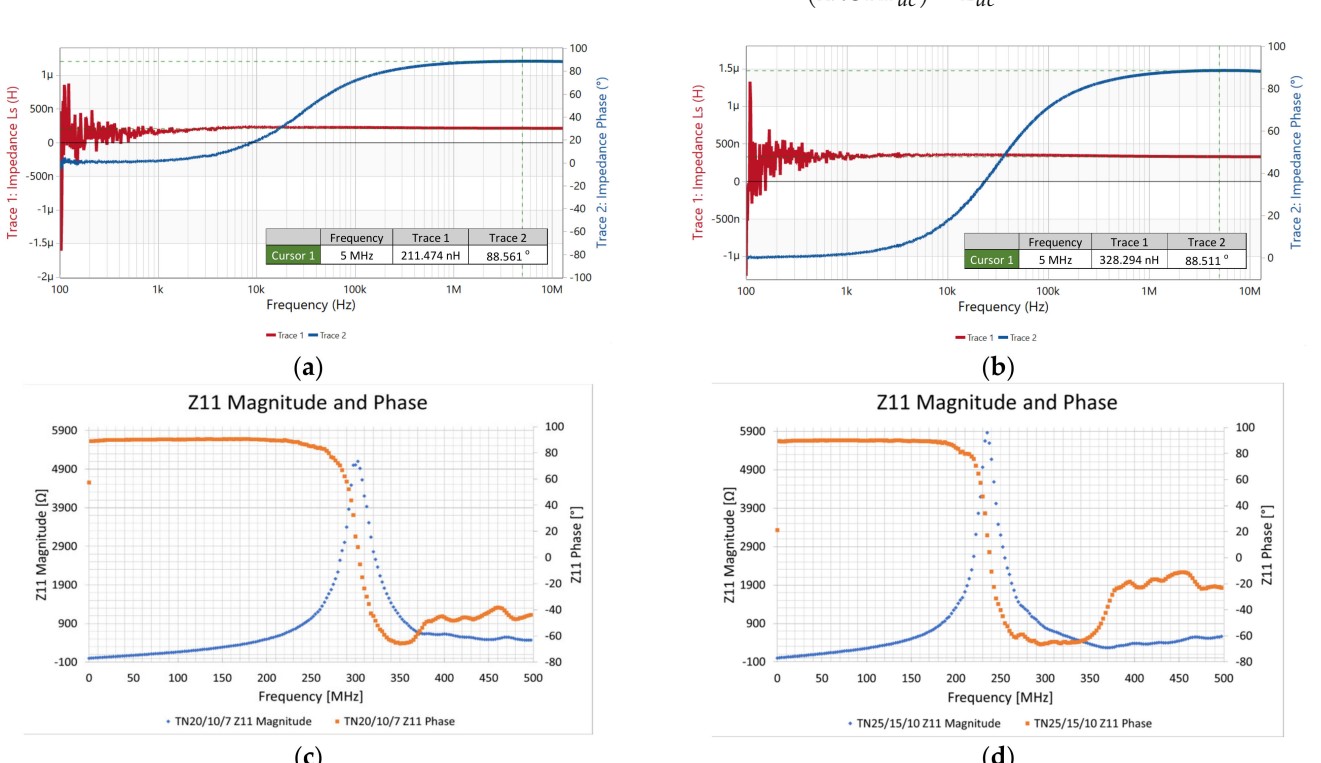

**Figure 16.** Sample 2 and sample 3 width of ABS dummy core. (**a**) Sample 2 inductance value measured at 5 MHz: red—inductance value; blue—impedance phase; (**b**) sample 3 inductance value measured at 5 MHz: red—inductance value; blue—impedance phase; (**c**) sample 2 imaginary part of $Z_{11}$ impedance with visible resonant transition: blue—impedance value; orange—impedance phase; (**d**) sample 3 imaginary part of $Z_{11}$ impedance with visible resonant transition: blue—impedance value; orange—impedance phase.

The empirical results show that the winding capacitance wound on the non-magnetic dummy core is considered an interwinding capacitance between the wire windings themselves (Equation (27)); however, the slight increase in the capacitance due to the presence

of a core material with a certain permittivity (Equation (30)) must be taken into account. Moreover, the results show that the method given in [2] yields consistent results, with the relative discrepancy between $C_p$ (Table 1) and $C_{swc}$ (Table 2) being 28.2% for sample 2 and 4.4% for sample 3. Even better results are obtained for the inductors with the dummy cores, with the discrepancy between $C_{sdc}$ and $C_{sdc(meas)}$ being 15.5% for sample 2 and 4.3% for sample 3 (Table 2).

The estimation of the inductor stray capacitance and self-inductance has also been made by the FEM simulation carried out in Ansys Q3D. In the simulation, the frequency range has been set from 100 Hz to 6 MHz, which is twice beyond the self-resonant frequency of sample 2 and sample 3. The permeability, permittivity, and conductivity for both 3C90 and ABS cores have been assumed to be constant to exclude any possible natural or dimensional core effects. The values of the inductors' inductance and stray capacitance have been measured at 10 kHz in the case of the ferrite core and at 5 MHz in the case of the ABS core, which, as previously assumed, is the inductors' inductance plateau region. Additionally, the 1 mm spacing between windings and the core has been added for both samples to account for copper wire stiffness and its bending curvature around the core (Figure 5a,b).

The results show (Table 3, Figure 17a–f) that the discrepancy between the theoretical prediction of the windings' stray capacitance if a dummy core is used $\left(C_{s_{cd}}\right)$ and the FEM simulation ($C_{sFEM}$) is almost negligible, with a relative error of 0.9% for sample 2 and 1.53% for sample 3. Moreover, the simulation shows that the 3C90 magnetic core with a high permittivity of $10^5$, high permeability of 2400, and relatively high conductivity of 4 has only a negligible impact on the windings' capacitance, increasing its value only by 17.3% for sample 2 and by 17.6% for sample 3. This suggests that the resonant frequency seen in the impedance characteristics of sample 2 and sample 3 (3C90 core based), and thus the overall equivalent stray capacitance of the inductor, is mostly due to the dimensional resonance of the core and associated with these phenomena.

**Table 3.** Ansys Q3D FEM simulation parameters and results for sample 2 and sample 3.

| Sample No. | Core Material | MLT [mm] | $d_i$ [mm] | N [-] | $\varepsilon_r$ [-] | $\mu_r$ [-] | $\sigma$ [S/m] | $L_{(FEM]}$ [nH] | $C_{s(FEM)}$ [pF] |
|---|---|---|---|---|---|---|---|---|---|
| Sample 2 | ABS | 34.5 | 0.75 | 10 | 4 | 1 | $6.25 \times 10^{-18}$ | 257.3 | 1.10 |
| Sample 3 | ABS | 42.0 | 0.75 | 10 | 4 | 1 | $6.25 \times 10^{-18}$ | 290.7 | 1.31 |
| Sample 2 | 3C90 | 34.5 | 0.75 | 10 | $10^5$ | 2400 | 4.00 | $228.9 \times 10^3$ | 1.29 |
| Sample 3 | 3C90 | 42.0 | 0.75 | 10 | $10^5$ | 2400 | 4.00 | $253.7 \times 10^3$ | 1.54 |

### 2.3. Improved Equivalent Inductor Model

As mentioned, the inductor self-resonant frequency depends not only on the windings' capacitance but also on the apparent capacitance of the core.

In the case of the windings' capacitance, it is assumed that in certain inductor configurations, the ferrite core has only a negligible impact on it. Therefore, this capacitance is mostly defined by the coreless turn-to-turn capacitance ($C_{s_{nc}}$) and the capacitance of the core coating or bobbin ($C_{s_{wc}}$) (Equations (26)–(30)) [2].

In the case of the core, the capacitance exists between core crystal grains due to the high-resistivity phase deposited on the grain boundaries. This causes a high effective permittivity [26], which, when combined with ferrite high permeability, significantly reduces the length of the electromagnetic waves propagating through the core. At certain frequencies, the electromagnetic wave will resonate within the core, triggering dimensional resonance, and thus causing a sharp drop in the real part of the complex permeability. We might call the core capacitance an apparent one because it is not due to the charge storage but rather due to the complex resonance-based phenomena, which causes a winding–core flux decay. Its ESR is also a product of complex phenomena inside the ferrite crystal structure.

Nevertheless, to properly model the inductor, its stray capacitance shall be split into two parts, one related to the windings and one related to the core (Figure 18).

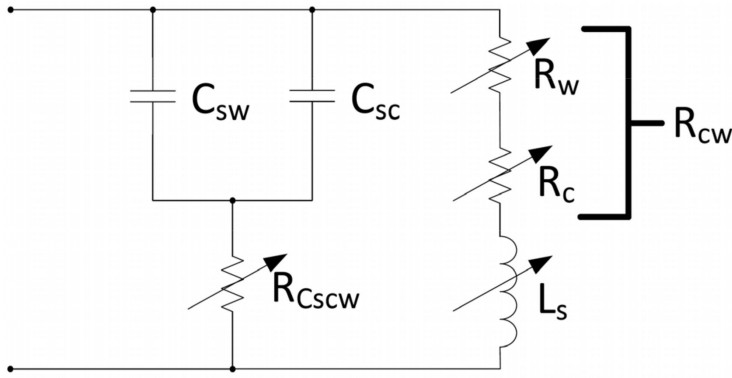

**Figure 17.** Ansys Q3D FEM RLC parasitic extraction of sample 2 and sample 3. (**a**) Sample 2 3D model with mesh shown; (**b**) sample 3 3D model with mesh shown; (**c**) sample 2 L-C characteristics with dummy core: red—stray capacitance; green—inductance; (**d**) sample 3 L-C characteristics with dummy core: red—stray capacitance; green—inductance; (**e**) sample 2 L-C characteristics with 3C90 core: red—stray capacitance; green—inductance; (**f**) sample 3 L-C characteristics with 3C90 core: red—stray capacitance; green—inductance.

**Figure 18.** Improved series equivalent inductor model.

Unfortunately, the relationship between the core and the windings is complex and implicit, and thus, there is not enough scientific evidence as to what is the correct relationship between their ESRs. To keep the model consistent with the obtained data, both ESRs have been combined into one resistance ($R_{Cscw}$).

The windings' capacitance ($C_{sw}$) can be estimated using Equations (26)–(30). The core capacitance ($C_{sc}$) can be estimated by simply subtracting the capacitance obtained from the L-C product at the inductor self-resonant frequency (Table 1). The combined ESR ($R_{Cscw}$) of the windings and the core can be obtained using Equations (17)–(22) and iterative approximation curve-fitting technique until the relative fitting error is small (e.g., less than 2%) along the entire frequency range.

## 3. Discussion

As stated, several existing complex permeability measurement and calculation techniques do not fully reveal how the complex permeability is measured and fitted, or the fitting is oversimplified [4–10,12] or overcomplicated [11]. Moreover, these techniques are often based on mathematical approximations and incomplete equivalent inductor models where several fitting variables fitted at once take any arbitrary values [12,13]. This approach cannot lead to valid results and does not represent the physical behavior of the inductor needed for SPICE-based simulation software commonly used in the industry [14,15].

The proposed method addresses this issue by implementing the iterative approximation curve-fitting technique. This technique is based on the inductor series equivalent model, with only one fitting variable, assuming that the inductance and the other quantities change along the frequency range according to the measured inductor impedance. This change and the model represent the physical behavior of an inductor, and only one fitting variable assures high fitting accuracy without the possibility of taking any arbitrary values by other quantities, which would otherwise be fitted. This allows for the accurate estimation of complex permeability regardless of inductor size, shape, winding structure, or frequency range. If the inductor stray capacitor ESR loss tangent is correctly defined (in case of sample 2 and sample 3 is set above 2), then it is possible to obtain a relative fitting error less than 2% along the entire frequency range (Figure 19a–d).

This careful study shows that the change in inductor inductance and other quantities is mostly due to the core's dimensional resonance and the windings' turn-to-turn and turn-to-core capacitance. The dimensional resonance also influences the inductor stray capacitance and its ESR values. The mechanism of how they originate is further investigated by test-bench measurements, calculations, and the FEM simulation, which are shown to be in good agreement. This has resulted in the proposal of an improved inductor equivalent model, which splits the stray capacitance into two parts: one related to the windings and one related to the core and frequency-dependent ESR of the stray capacitor.

However, the proposed techniques shall be further examined, especially if we consider multi-winding, multi-layer inductors with a significant stray capacitance related to the windings. Furthermore, the FEM simulation shall be improved using models with PA11 ferrite core coating and a close winding alignment to the core. This might prove with higher precision the capacitance relationship between the core and the windings, allowing for further verification of stray capacitance and its ESR values.

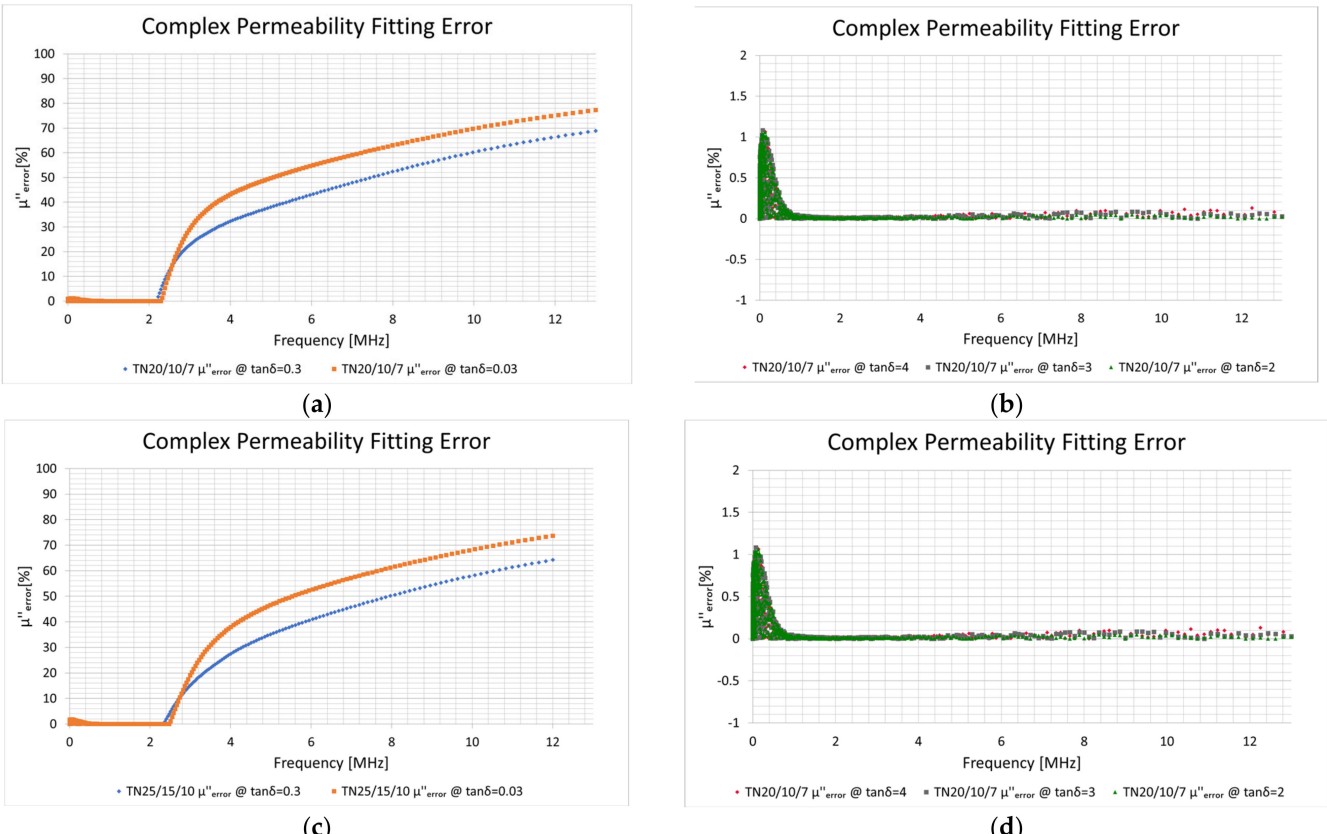

**Figure 19.** Relative fitting error between $\mu''_{r_{measured}}$ and $\mu''_{r_{estimated}}$ (**a**) sample 1: orange—$tand\delta = 0.03$; blue $tand\delta = 0.3$; (**b**) sample 1: green—$tand\delta = 2$; grey—$tand\delta = 3$; red—$tand\delta = 4$; (**c**) sample 2: orange—$tand\delta = 0.03$; blue $tand\delta = 0.3$; (**d**) sample 2: green—$tand\delta = 2$; grey—$tand\delta = 3$; red—$tand\delta = 4$.

## 4. Conclusions

This paper shows improved measurement and calculation techniques regarding the complex permeability of ferrites, one of the essential quantitates that define inductor behavior in the frequency domain. This is significant because it might help in the development of improved inductor loss models and universal simulation models (e.g., SPICE models) that capture all AC loss mechanisms (core loss, winding loss, etc.), which do not yet exist, especially if we consider high-frequency power electronics applications.

The proposed method is based on the inductor equivalent model, and thus allows for the correct estimation of complex permeability values directly from the impedance measurement regardless of inductor size, shape, winding structure, or frequency range. As the results show, it reveals with high accuracy not only the low-frequency complex permeability values, e.g., the values of the imaginary part stripped from the influence of windings' resistance, but also the permeability peak values existing in the vicinity of the resonance. This information would be otherwise hidden or diminished by parasite(14ic components, of which the real inductor is made.

The method is also simple and intuitive to process, assumes the frequency dependence of most of the inductor model components, and thus overcomes some limitations and complexity of other methods.

**Author Contributions:** Conceptualization, P.S.; methodology, P.S.; software, P.S.; validation, P.S., S.L., and C.W.; formal analysis, P.S.; investigation, P.S.; resources, P.S.; data curation, P.S.; writing—original draft preparation, P.S.; writing—review and editing, P.S., S.L., and C.W.; visualization, P.S.; supervision, S.L. and C.W.; project administration, P.S., S.L., and C.W.; funding acquisition, S.L. and C.W. All authors have read and agreed to the published version of the manuscript.

**Funding:** This research was funded by Fideltronik Poland Sp. z o. o. and by the AGH University of Krakow, Faculty of Computer Science, Electronics and Telecommunications, Institute of Electronics under a joint implementation doctorate scheme.

**Data Availability Statement:** No new data were created or analyzed in this study. Data sharing is not applicable to this article.

**Acknowledgments:** The authors would like to thank the management board of Fideltronik Poland Sp. z o. o. for continuous support of this work.

**Conflicts of Interest:** The authors declare no conflict of interest.

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
