# Peer review of "Measurement and Calculation Techniques of Complex Permeability Applied to Mn-Zn Ferrites Based on Iterative Approximation Curve Fitting and Modified Equivalent Inductor Model"

_electronics, doi:10.3390/electronics12194002_

Round 1

Reviewer 1 Report

Measurement and calculation techniques of complex permeability applied to Mn-Zn ferrites is proposed in this manuscript. In my opinion, the manuscript should be modified thoroughly. The following suggestions can be considered by the authors to improve the quality of the present manuscript:

1 The quantitative accuracy of the proposed model can not be found at the Abstract and Conclusion section.

2. There exists some errors of English expressions and please modify the whole manuscript carefully.

Good

Reviewer 2 Report

1. The title is relatively far from the main paper's contribution (it does not clearly reflect the purpose of the paper): to be adjusted

2. In the abstract, the sentence "This is only...get it right" is to be reformulated. It is required to also briefly indicate what is "complex permeability" in a way that readers can directly understand the paper's aim

3. The English in the introduction is bad

4. To clearly identify each abbreviation upon first usage

5. A reference is needed for "impedance analyzer" upon first usage

6. A reference is needed for the "Dowell's equation" upon first usage

7. Figure 1 hasn't got anything to do with the ESR identification

8. A reference is needed for "measurement uncertainties" upon first usage. In fact, the entire introduction section is almost condensed information without a proper referencing nor structuring: this issue must be well addressed, with a clearer literature review fragment

9. In section 2, the sentence "Moreover,...extended" is to be completely removed or reformulated with a proper reference

10. It is difficult to understand the relation between the inductor's lumped model and "complex permeability" values: to give a clear link between the two stated entities

11. All figures should be properly in-text cited, rather than the used method 

12. It is unclear why the Figure 2-(f) is inserted among the inductor's different models

13. As the case was for Figure 2, Figure 3 is also irrelevant

14. Through the mathematical modelling from Eq.(1) to Eq.(11), exist many variables/constants that are left undefined

15. Table 1 is inappropriately inserted 

16. You stated how to calculate the "fringing factor" without a previous usage or a proper explanation: how does such a factor fit into the revealed mathematical model from Eq.(1) to Eq.(11)?

17. Figure 4: is inappropriately inserted in the manuscript + contains unclear data + needs polishing + title restructuring (it can be decomposed into multiple images when not applicable)

18. What is the link between sections 1.1 and 1.2? How did you move from the inductor's lumped model directly to verifying the "complex permeability" values?

19. It is a maze-like seeing two figures (i.e., 2f and 3) and the same time to understand a single idea

20. The symbols "core cross-section area" to "winding wire diameter" are inappropriately inserted in the manuscript. It takes time, effort, and patience for the reader to know that such abbreviations are in accordance with Table 1

21. Figure 6 is literally unreadable where it is not cited along with Figure 5 and too many other figures also!

22. Some figures' titles are larger than some of the manuscript's fragments!

23. "loss tangent of PA11 to 3C90": what are PA11 and 3C90?!

24. Sub-sections from 2.2.1 to 2.2.5 are difficult to follow/link

25. What does "complex permeability" has to do with your suggested improved inductor model? In Figure 18 (the supposed improved inductor model), I can only see a replication of Figure 2-(b): to justify!

26. The "Discussion" part is more a self-assessment of the manuscript! It hasn't got anything to do with an actual numerical/tabulated/graph-based critical investigation of the paper's outcomes

27. There is no clear scientific significance in the "Conclusion" part. To not insert any references in the conclusion, and better indicate concrete outcomes from the findings.

-------------------

The English of the entire manuscript is to be word-wise scanned

Each figure's quality is to be polished

The paper needs real attention, since its different sections are perceived not properly linked

The outcomes are not significantly proofed

Referencing is suggested to be in chronological order

english is poor

Reviewer 3 Report

The paper describes a useful practical method for determining the complex permeability of magnetic cores of inductances. The technique is particularly useful for applications in high frequency power electronics where simple impedance analysis based on small-signal probing fails. With improved modeling of the inductor for both the cases with and without air gap in the core, updated equivalent circuits are established, their transfer functions are calculated analytically and a methodology is established for extracting from the impedance measurement not only the complex permeability values but also the losses in core and windings.

To the best of my knowledge, the work is novel and original. The title of the paper is appropriate. The abstract summarizes the approach and covers the main aspects of the work. The paper is well structured. In the introduction, the work is put into proper context to previously published research. Several equivalent circuit models of the inductor are presented, in conjunction with their analytical description of real and imaginary parts of the core permeability. The figures and illustrations are clear and facilitate comprehending the key messages of the paper. With the examples of six sample inductors, the methodology of extracting the complex permeability from impedance analyzer measurements by iterative curve fitting is shown. The importance of good estimates of the loss tangent is illustrated. Determination of the capacitance contributions is performed, with distinction between the capacitance between adjacent turns and the capacitance to core. In the conclusion, the main findings are summarized.

The following points should be addressed in revision:

1. The paper is very lengthy. In my opinion, it would gain in readability and acceptance if it were somewhat shortened. Let me mention just a few examples what could be omitted in my opinion: the equivalent circuit in Figure 2a, with R_leak and L_leak, is not used further on in the paper. Then serial and parallel models for ungapped and gapped cases could be arranged in Figure 2 as a 2 × 2 matrix. The equations (2) and (3) for real and imaginary parts in ungapped case are the same as equations (6) and (7) in gapped case (just replace R_cw with R_cwg). The same is true for equations (10) and (11), being repetitions of (8) and (9), only with index g for gap added. Figure 4 could be unified if all real curves were shown in blue and all imaginary in red, then the caption could simply read: “Effective complex permeability plot of (a) sample 1 to (f) sample (6), with real part shown in blue and imaginary part in red.” To facilitate, I would simply write the sample number in the graphs 2a .. 2f. In my opinion, it is nice to visualize the loss tangent fitting in figures 9, 10 and 11, but it could also be shown for just one sample, with all the curves for different tan delta plotted in one graph, maybe with separated graphs for real and imaginary parts to improve legibility. I suggest that the authors carefully check their whole paper for redundancies and for possibilities to shorten it, of course without compromising content.

2. A references should be given for the less commonly known term “Dowell’s equation” (line 40), particularly since the equation is not given in the paper.

3. Some acronyms have not been defined, for instance “equivalent series resistance (ESR)” in line 38 and “finite element method (FEM)” in line 502.

4. Line 160: add the manufacturer: “Bode 100 impedance analyzer from “Omicron Lab”.

5. Line 14: I suggest to replace the colloquial term “far from great” by somewhat more professional wording, for instance “less precise” or “slowly converging” or “ill-posed”, and instead of “to get it right” in the next line, rather write something like “to achieve convergence”.

6. Typo: the past tense of “to wind” is “wound” (line 238).

Round 2

Reviewer 2 Report

The quality of the paper has been significantly improved according with the raised comments. I recommend it from my side to acceptance.